# Gender perspectives on zoonotic disease epidemiology; A strength weakness opportunities threats analysis in Bundibugyo district, Uganda

**Clovice Kankya[1]*, James Muleme[1,2], Lydia Nabawanuka Namakula[2], George Seruwagi[1,3], Christine Mbabazi Mpyangu[4], Lesley Rose Ninsiima[1,2]**

**1** Department of Biosecurity, Ecosystems and Veterinary Public Health, College of Veterinary Medicine Animal Resources and Biosecurity, Makerere University, Kampala, Uganda, **2** Department of Disease Control and Environmental Health, School of Public Health, College of Health Sciences, Makerere University, Kampala, Uganda, **3** Department of Mathematics, School of Physical Sciences, College of Natural Sciences, Makerere University, Kampala, Uganda, **4** Department of Religion and Peace studies, School of Liberal and Performing Arts, College of Humanities and Social Sciences, Makerere University, Kampala, Uganda

* clokankya@gmail.com












## Abstract

### Background

Gender mainstreaming in zoonotic disease prevention and control is paramount to produce sustainable solutions as well as keeping communities at the human-animal-environment interface safe and healthy. It is important to note that zoonoses register high mortality rates globally once they occur and they are highly transmissible. Hence this study aimed to explore the gender perspectives on zoonotic disease epidemiology using a (strength, weakness, opportunities and threats) SWOT analysis in Bundibugyo district, Uganda.

### Methods

This study employed a descriptive participatory approach, utilizing data gathered sequentially from 12 Key informant interviews, 4 In-depth interviews and 4 Focus group discussions methods in Bundibugyo district. Interviews used interdisciplinary groups systematically using a SWOT analysis. The research methodology employed an interpretative phenomenological analysis (IPA) approach allowing participants to articulate their perspectives in their own words. Data analysis was done using both inductive and deductive thematically using NVIVO 12 pro, facilitating the development of codes, sub-themes, and themes.

### Results

The results of this study prioritized key zoonotic diseases within the district that affect community health. The key themes from these results a) Threats increasing risk of

**Data availability statement:** Data cannot be shared publicly because it contains potentially identifying and sensitive survivor information. Data restrictions have been imposed by Makerere University Research Ethics Committee, School of Health Sciences Institutional Review Board and CIDIMOH Project. The data underlying the results presented in the study are available from the CIDIMOH secretariat, who can be contacted via cidimohsec@mak.ac.ug for researchers who meet the criteria for access to confidential data.

**Funding:** We acknowledge funding obtained from Norwegian Agency for Development Cooperation (NORAD) through the NORHED-II Program and the project Climate Change and Infectious Diseases - A One Health Approach (CIDIMOH), grant number 68802. The funders had no role in study design, data collection and analysis, decision to publish, or preparation of the manuscript.

**Competing interests:** The authors have declared that no competing interests exist.

zoonotic spillover, b) Community weaknesses increasing zoonotic diseases occurrence, c) Community strength for zoonotic disease management, prevention and control, d) Opportunities for communities in management of zoonotic diseases. The study emphasizes that political instability, land migration, food insecurity, cultural hunting practices, and climate change act to increase the risk of zoonotic diseases. Complications arise from the mis-identification of diseases due to similar symptoms, and a lack of community education about these diseases. The risk of exposure is influenced by gender roles, with men, who generally interact more with animals, at higher risk. Conversely, women, due to their roles in caring for the sick and involvement in child immunization, are at risk but also play a crucial role in disease control. Despite these challenges, there are opportunities for disease management and prevention such as leveraging experienced health workers for disease identification and education, utilizing local communication channels, engaging opinion leaders for effective risk communication, and providing regular training for health workers could address these issues. However, limited funding hinders the execution of recommended strategies such as regular surveillance, tracing of suspected cases, and health register reviews.

## Conclusion

This study emphasizes the necessity of gender-sensitive approaches in understanding and mitigating zoonotic diseases, advocating for strategies that recognize socio-cultural factors, promote health education, and tailor interventions to provide comprehensive care and protection for all, irrespective of gender.

## Introduction

Globally, there has been an increase in human-animal interactions for both social and economic reasons, leading to an increase of zoonotic infections over the past years [1]. The increase in the burden of zoonotic infections is threatening the world population due to the associated widespread geographical dispersion [2]. The emergence and re-emergence of zoonotic diseases such as Rift Valley fever, highly pathogenic avian influenza (HPAI), Marburg virus disease (MVD), Ebola virus disease (EVD), Brucellosis and Tuberculosis are indicative examples of the interconnectedness between the humans and animals (wildlife and domestic) [3]. Community desires and endeavors to feed and survive, have also contributed to this interaction and thus related health consequences [4]. For instance, World Health Organization (WHO) case fatality rates for zoonotic illnesses like EVD and MVD to range from 24 to 88 percent [5] and 25–90 percent [6], respectively. In addition, over 63 million livestock farmers (both women and men) around the world are susceptible to bacillus anthracis and an estimated 100,000 cases of anthrax worldwide are recorded each year [7]. Several factors have been epidemiologically linked to zoonotic disease transmission chain and subsequent management. However, social cultural aspects including gender are gaining international scientific recognition especially in low resource settings

such as forest and game communities [8]. According to the WHO, gender refers to characteristics and roles of women, men, girls and boys that are socially constructed [9]. For instance during the 2014 EVD outbreak of Liberia, West Africa, 75% of the task force was comprised of women [10]. In addition, women have a socio-cultural obligation taking care for the sick both at home and in medical facilities. On the other hand, some men and boys have been involved in hunting for bats, calabash monkeys among others where they may be exposed to various zoonotic pathogens thus culminating into index cases in certain reported outbreaks [11]. These eventually spread the infection to other close contacts through several cascade events [12,13].

Uganda has reported a number of zoonotic disease outbreaks, including Crimean-Congo hemorrhagic fever, anthrax, MVD, and EVD with the index cases originating from the zoonotic disease hotspots [14]. Hotspot areas in Uganda that have experienced such outbreaks include the cattle corridors of Uganda, the Uganda-Kenya border in the Kween and Kapchorwa districts [15], and Bundibugyo district [16] among others. Uganda battled a Sudan Virus - Ebola outbreak between September 2022 and January 2023 in the districts of Mubende and Kassanda [17]. In 2007, the Ugandan Ministry of Health (MoH) officially announced an outbreak of EVD in Bundibugyo district which was managed at Kikyo Health Centre IV in Bughendera County and at Bundibugyo General Hospital [14]. These zoonoses have led to significant economic losses and human fatalities globally, not just in Uganda. The implemented prevention and control measures have instilled fear in tourists, reduced government revenue, led to school closures, and overwhelmed healthcare systems [18]. A strong emphasis on the integration of sex and gender perspectives in zoonotic disease research and management could help stakeholders and policy makers to conceptualize transformative solutions for most burdened populations especially in the affected areas [19].

Gender roles lead to unique health risks for men and women by altering their exposure to pets, wildlife, and the environment [20]. Understanding and addressing these differences are vital for effective zoonotic disease prevention and response strategies [21]. In order to identify the "at-risk" cultural practices and roles, and to prevent current epidemics, it is worth investigating the gender roles and cultural orientation of people in these zoonotic disease "hotspots" in order to contribute to the achievement of the Sustainable Development Goal 3: Good Health and Wellbeing for all, regardless of peoples' sex and gender [3]. The aim of this study, therefore, was to explore the gender perspectives on zoonotic disease epidemiology using a (strength, weakness, opportunities and threats) SWOT analysis in Bundibugyo district, Uganda.

## Materials and methods

### Study design

The study utilized descriptive participatory epidemiology to generate sequential data through different qualitative approaches providing equal priority to different genders. This study involved 12 key Informant interviews (KIIs), 4 In-depth interviews (IDIs) and 4 Focus group discussions (FGDs) with pretested respective qualitative data collection tools to guide community interactions between 23rd October 2023–30th November 2023.

### Study site

This study was conducted in Bundibugyo district. Bundibugyo district is located in the Western region of Uganda and bordered by Democratic Republic of Congo (DRC) in the West, Ntoroko district in the Northeast, Bunyangabu district in the Southeast and Kasese district in the South [22]. The geographic boundaries of the district are Semliki River to the west, Rwenzori mountains to the east, and Lake Albert to the north [5,17]. Thenational population census put the district's population at 263,800 with a 309.8/km² population density and 2.8% annual population change from 2015 to 2022 [22,23].

Bundibugyo district was selected because of its geographical (mountainous and boarding Congo basin) and ecological characteristics in relation to the numerous zoonotic disease outbreaks within the region [24]. Bundibugyo district has recorded one of the worst and longest EVD outbreaks in Uganda with a very high case fatality rate ranging from 50% to

90% [25]. All the 7 sub counties (Bubandi, Bubukwanga, Bundibugyo Trading center, Busaru, Harugali, Kasitu, Nduguto) were considered and represented due to their prior experience with zoonotic disease outbreaks.

### Sampling and recruitment

Participants for the study were recruited basing on their prior experience with zoonotic disease outbreaks and management using a snowballing sampling method. Participants for the KIIs were recruited from, animal health, human health, environmental and public health, community development. The FGDs participants were recruited from elders and village opinion leaders, Village Health Teams, Community animal health workers and community members while IDIs were recruited from Ebola survivors and previous cases of zoonotic disease infections (tuberculosis and brucellosis). The FGDs were stratified according to females and males. We recruited 40 participants for the FGDs from those who qualified to participate with 16 females and 24 males. All participants ranged from 18 years and above and understood English.

### Study participants

The following Table 1 indicates the study participants who provided input in line with their professional expertise, lived experiences and local perspectives.

### Data collection

The qualitative data collection guides were first pretested in Kasese district with similar characteristics to the study site, where the participant's gender perspectives on zoonotic disease transmission, occurrence and management were captured. The revised and pre-tested discussion guides were structured around five key areas, including gender roles in the transmission of zoonotic diseases, the differential risk exposure among genders in zoonotic disease outbreaks, and strategies for district/community prevention of zoonotic disease outbreaks. Additionally, sections addressing soft skills such as community entry, rapport building, participant selection, facilitation, and documentation were thoroughly discussed.

For IDIs and KIIs, respondents discussed and reported the different gender roles socially constructed in their communities, and identified factors related to exposure, occurrence, and management of most common zoonotic diseases within their communities.

For the FGDs, each participatory session was limited to 6–8 participants making it a total of 4 sessions with 2 for females and 2 for males, to allow for an adequate level of interaction from all participants during group activities and discussions. Participants were each given a code in the form (sex## ~ where ## is a number) for gender disaggregated data. Letters W and M were used to refer to a woman and man respectively. For example, the first participant was identified as W01 if she was a female or M01 if he was a male.

Briefly, for each interview (KIIs, IDIs and FGDs), we used the SWOT analysis tool in identifying gender roles in relation to zoonotic diseases, exposures of different genders to these diseases, opportunities to prevent outbreaks and exposures to zoonotic diseases as well as identifying challenges in the prevention of zoonotic diseases. All interviews are carried out in English as most people involved in this study knew English and there was not need to use the local language.

**Table 1. Disaggregated number of participants in the study.**

| Interviews | Number of participants | |
|---|---|---|
| | Females | Males |
| Key Informant Interviews | 7 | 5 |
| Focus Group discussions | 16 (2 female groups) | 24 (2 male groups) |
| In-depth interviews | 2 | 2 |


### Data analysis

The research methodology employed an interpretative phenomenological analysis (IPA) approach, chosen for its suitability in exploratory studies with small sample sizes, allowing participants to articulate their perspectives in their own words. Audio recordings were transcribed verbatim and shared with participants for validation of accuracy before being entered into NVivo 12 pro software for analysis. The data analysis process adhered to the flexible IPA guidelines, which involved multiple readings of transcripts and the generation of researcher notes containing thoughts, observations, and reflections. These data were subsequently transformed into emerging themes, with a focus on identifying relationships and clustering themes. Two members of the research team (LRN and LNN) meticulously read and coded all interview transcripts, (CK) resolving any discrepancies or concerns through discussion until consensus was reached on a set of themes.

### Ethical consideration

Ethical approval was sought from Makerere University with reference number (MAKSHSREC-2023–573). Scientific and ethical clearance to conduct this study was obtained from the Uganda National Council for Science and Technology (UNCST) with reference number (HS3297ES). The study was reviewed by the research ethics committee and found to be scientifically and ethically satisfactory, and it was approved. In addition, permission and clearance was obtained to conduct the study from Bundibugyo district authorities. Participants gave written informed consent using a consent form that was signed during data collection.

## Results

Our results revealed differences and inequalities in the gender division of labor, access to and control over resources and a general lack of knowledge on zoonotic diseases among women and men. The key themes from these results included a) Threats increasing risk of zoonotic spillover, b) Community weaknesses increasing zoonotic diseases occurrence, c) Community strength for zoonotic disease management, prevention and control, d) Opportunities for communities in management of zoonotic diseases.

### Listing of priority Zoonotic diseases in Bundibugyo district

Participants from KIIs, FGDs and IDIs actively mentioned their understanding and listed key zoonotic diseases that affect the area. A key number of zoonotic diseases were mentioned by all participants such as, Ebola Virus Disease (EVD), Marburg Virus Disease, Tuberculosis, Rabies, Anthrax, Brucellosis, Rift Valley Fever. However, the priorities among the zoonotic diseases included Ebola virus, Marburg virus, Anthrax and Tuberculosis were prioritized by participants. All these formulated lists were merged and only those common among the different participants were prioritized. Thereafter, participants were requested to generate the local names of the prioritized zoonoses, which were also later harmonized with cultural leaders within the study area (Table 2) however, some did not have the local names as they are known by their English names and were not included in this table. It is important to note that zoonotic diseases such as EVD and MVD did not have a local name associated with them.

**Table 2. Showing the different zoonotic diseases known and their local names.**

| Zoonotic diseases | Local names |
|---|---|
| Tuberculosis (TB) | *Akakongo* |
| Brucella | *Omutsutsa webhisoro* |
| Anthrax | *Akasusu* |


### Threats increasing risk of zoonotic spillover

FGD participants reported that one of the reasons for land encroachment was political instability in the nearby districts which forces land migration on informal settlement and food insecurity. Due to food insecurity in the communities, some have resorted to hunting wild game meat as a source of food. In addition, some communities in Bundibugyo district find hunting a cultural acceptable practice. Hunting was reported to be one of the major contributing factors to zoonotic outbreaks in communities. This was reported to be an activity mainly performed by men and male children within a community. Among the hunted animals included monkeys, birds and bats. Hunting is also done simultaneously during domestic animal grazing time of the day. Therefore, men and male children's interaction with wild animals increases their exposure to zoonotic infections.

> *"There are communities that stay high in the mountains and those that stay near game parks that usually hunt wildlife as a source of food or revenge since the wildlife destroy their crops therefore once they have access to it, they kill the animal and feed on it. This is usually done by men"* (KII-10).

Encroachment on different game reserves and national parks, exacerbated by the impacts of climate change has led to increased exposure to zoonotic diseases. This has not only threated biodiversity but also destabilizes ecosystems and increases the risk of human wildlife conflict. The activities of animal slaughter and subsequent meat transportation, typically carried out by men, was highlighted by participants to potentially put them at a higher risk of exposure to various zoonotic diseases. These activities often involve close physical contact with animals or animal products which may be infected. By the time inspection occurs, initial exposure might have already occurred, underscoring the importance of implementing safety measures like the use of personal protective equipment, good hygiene practices, and immediate reporting of ill animals.

> *"The national game parks have been encroached by different people and it has become difficult to control game meat hunting and causing climate season changing. The men continue to do the hunting and, in most cases, in case any wildlife has a disease that is zoonotic then its them affected yet they may end up not going to the hospital"* (KII-04).

### Community weaknesses increasing zoonotic diseases occurrence

Women highlighted to be responsible for the general cleanliness of a homestead. Women and youth are involved in various household activities that ensure a clean home such as laundry, washing dishes and kitchen ware, food preparation and all household chores that ensure sanitation and hygiene. The participants highlighted that it was rare though possible for a man, the head of a family, to be engaging in such activities. The possibility depends on individual views of the man's role to support their spouse in household chores and how they socialize within the community. Men's involvement in sanitation and hygiene were reported to rotate around ensuring safety and hygiene of their animals which is connected to the economic gain associated with animals. Men together with the male children engage in the cleanliness of animal houses, feeding troughs and general welfare for the domesticated animals. This implies that men's roles often include animals, hence being at higher risk of zoonotic exposure compared to women. During the FGDs, one of the female participants said,

> *"After work, I have to go home and cook for my family dinner and also clean every morning, but the men here just go back home to check how animals have been managed within the day and then eat food. If the disease is within the animals, it means that the men will be the first to get the disease"* (W03).

Culture, in terms of food, encompasses the culinary practices, traditions, customs, and beliefs associated with the preparation, consumption, and sharing of food within a particular society or community. Culture is shaped by the types of

food that are deemed acceptable within society. Both male and female participants mentioned that certain tribes in the mountains have traditionally consumed bats as a regular meal for many years without experiencing any health issues. Bats are valued for their nutritional content and traditional medicinal properties. However, they have also been linked to diseases such as Marburg virus disease. Given that bats are consumed by men, women, and children within households, all members are equally exposed to the associated risks.

> *"Bats are a delicacy in the different homes in Bundibugyo district. I personally eat it and it's very healthy and has nutritious components such as iron for the body. Some women in the community use it for other purposes like witchcraft which shows how important they are in the community"* (M09)

Participants noted that people are not aware of the signs and symptoms of most of the zoonotic diseases which greatly affect the knowledge levels of the community. Some zoonotic diseases were presented to manifest in a similar way to other tropical diseases such as malaria, dysentery among others. In the in-depth interviews, participants exhibited diverse levels of awareness concerning zoonotic infections such as tuberculosis (TB), brucellosis, and anthrax, both in animals and humans. While some individuals demonstrated a comprehensive understanding of the clinical signs and diagnostic methods for detecting zoonotic diseases, others displayed limited knowledge, indicating gaps in awareness and education. Additionally, participants showed varying levels of awareness regarding the dangers of consuming raw milk, with some expressing heightened awareness of the risks associated with Brucella contamination and others demonstrating only a limited understanding of the potential health hazards. In addition, information on zoonotic disease outbreaks takes a long time to reach the wider community due to poor communication network and as a result, you find that the disease can easily disseminate to the wider communities.

> *"One time, I visited a friend who is of a certain tribe that drinks raw milk as a tradition, so as a visitor, I decided to try it but after a few days when I went to the hospital, the doctor diagnosed me with Brucellosis. There is a saying, "Taking raw milk is like playing Russian roulette with Brucella." I am now aware of the signs of brucellosis."* (IDI-03).

### Community strength for zoonotic disease management, prevention and control

Most participants identified the role of immunization at an early age, where children are still breastfeeding to be done by women because of their nurturing nature hence can afford to wait in the queues at the health facility for the immunization of their children like Tuberculosis. Most women in Africa have a role of caring for the sick. Women have been involved in various roles during caring for the sick such as washing their beddings, feeding those unable to eat on their own, cleaning vomits, bathing the sick among other roles. Men on the other hand care in monetary terms by providing food in the home that the sick will eat and finances for transportation to the health facility if required. This therefore keeps them physically away from the sick hence reducing their exposure to zoonotic diseases.

> *"As an Ebola survivor, my wife used to take care of me after I left the hospital, and she made sure I was well fed and washed the clothes and beddings to avoid the spread of the virus to our children"* (IDI-02).

Most participants revealed a range of perspectives on the dangers and high cases of anthrax among the community people. Preventing anthrax in livestock is the most important step in reducing the risk of human infection. While some individuals displayed a profound understanding of the potential risks associated with anthrax exposure and cited specific cases within their communities or regions.

> *"For us to be able to limit diseases such as anthrax in people, we need to emphasize the importance of livestock vaccination, especially in areas with high livestock-to-human populations or where anthrax is known to be naturally occurring."* (IDI-04).

### Opportunities for communities in management of zoonotic diseases

Experienced health workers could provide crucial education for clients waiting in facilities, sharing insights into zoonotic disease identification, prevention, and management. However, a key challenge remains the scarcity of such health workers. Therefore, strategies focused on scaling up training, mentorship programs, and knowledge sharing opportunities could help to build a larger workforce equipped to manage zoonotic diseases. The participants stated that it would strengthen our healthcare system's capacity not just to handle outbreaks when they occur, but also work towards preventing them. Not only can this improve the capacity of our health facilities to manage prospective zoonotic disease outbreaks, but it can also contribute to effective health education. This has ultimately trickled down to the communities to enhance community-based surveillance and reporting as well as prevention of zoonoses spread and spillover.

> *"We have health workers who managed zoonotic diseases in the past outbreaks. These know which signs to look out for in a patient. That is a strength that we can use to improve our health facilities in managing future zoonotic diseases. They can also health educate clients as they come in the waiting room although these workers are few"* (M08).

Participants recommended the use of locally available communication channels like radios, public audio speakers, religious leaders to reach the physically distant in the mountains. Talk shows on the other hand were reported to be important in lobbying for funds in regard to zoonotic diseases management and control.

> *"It would be a good thing to use our community radio to pass on health talks about zoonotic diseases. Especially people here in the town areas, they will listen in and at least know what to do in the face of similar signs and symptoms to zoonotic diseases."* (W02).

Effective risk communication ensures compliance to communities in regarding zoonotic diseases management. Opinion leaders including religious leaders, Village Health Teams among other stakeholders are always in liaison with the community and their views have the power to influence the attitudes of communities towards or away from zoonotic diseases prevention strategies.

> *"There are people in this communities who speak and are listened to. In other words, they have influence over a large group of people. Those people should be talked to and sensitized to pass on information about zoonotic diseases and their prevention measures."* (KII-05).

More trainings refresher courses and zoonotic disease related Continuous Medical Educations (CMEs) were recommended to health workers serving in health facilities to build their capacity in early identification and diagnosis and management of suspects of zoonotic diseases such as EVD and MVD.

> *"The health workers in this district need more support to train them on the different zoonotic diseases. Fine we have had outbreaks before and they were key in diseases management, but they need to be remained so that they are alert in case of any zoonotic disease signs and symptoms present"* (KII-08).

Participants noted that food inspection should be one of the strategies done to prevent zoonotic diseases. Bundibugyo also has veterinary and environmental health officers, who in addition to the health workforce at health facilities, coordinate surveillance of zoonotic diseases. This should be done by professionals and carried out regularly by health inspectors and veterinary officers at the district to verify the safety of animals for slaughter and that of carcasses for consumption.

*"The health inspectors ideally should inspect the state of animals before slaughter and of the meat after slaughter. They know the signs of animal with bovine TB or even anthrax for example. A health inspector in liaison with the vet has a duty to ensure the safety of the public through animal and meat inspections."* (W06).

It was also emphasized that tracing of suspected cases, active surveillance, was often carried out during active outbreaks, although the scale was hindered by limited funding.

*"We do some surveillance though not at its best, but there is a lot of information from health registers about different conditions. That information when reviewed on a regular basis and summarized helps to understand the state of the district"* (KII-05).

## Discussion

Gender mainstreaming enables a better understanding of the varying impacts of infectious and emerging diseases and disorders at all levels of society. Women, men, girls and boys are viewed as people that are or can be infected and affected by zoonotic diseases in various ways. In this study, we delve into the intertwined dynamics of gender and zoonotic disease epidemiology in Bundibugyo district, Uganda, utilising a SWOT approach (Strengths, Weaknesses, Opportunities, and Threats). The goal is to capture an intricate, comprehensive picture of how diseases manifest and traverse along lines of gender.

Certainly, immunization and vaccination of children and animals serve as a significant protective shield against a range of zoonotic diseases. This practice is particularly important, as many of these diseases can be transmitted between animals and humans. It's crucial to notice the differential roles men and women have in healthcare, forming a dichotomy in their disease awareness levels [26]. From a gender perspective, it seems men predominantly take charge of animal vaccination in many communities, ensuring their livestock are protected against diseases [27]. In this study, men oversee animal vaccinations and engage directly with veterinarians a conduit to essential information about potential zoonoses including disease indicators in animals and various preventative measures. Conversely, women generally shoulder the responsibility of having their children immunized. This duty invariably educates them about a host of diseases and their symptoms, and the steps they can take to safeguard the health of their families [28]. This exposure to awareness information from vaccination and immunization activities empowers both women and men. Consequently, they are better prepared to access timely medical intervention during zoonotic disease outbreaks, which can significantly reduce morbidity and mortality [28].

Bundibugyo has had a history of zoonotic diseases outbreaks in the past which were detected and managed by health workers in the district along with external health teams [28]. The health workers interact with community members and provide health education sessions about zoonotic diseases. The interaction with the health professionals like the veterinarians and environmental health officers gives men an understanding of the different medications that the animals require including potential vaccines [29]. The presence of a one health team has increased capacity of related communities for early detection of zoonotic diseases. Timely reporting, response and monitoring risky activities in communities such as ante-mortem and post-mortem inspection of animals helps to deem animals fit for human consumption [30].

The weaknesses uncover the gender-based vulnerabilities to zoonotic diseases. Men face higher infection risks due to their extensive involvement in animal care. The sources of most of the infectious diseases, especially zoonotic diseases, have been associated with animal products and animal fecal waste. Some domestic animals such as pigs are reservoirs of the infectious diseases [31], which puts the males at more risk to these diseases. Women, shouldering the responsibility of caring for the sick, grapple with emotional and economic burdens, often exacerbated by a lack of psychological support and limited access to information [15]. Girls and women also carry most of the weight of the economic impact of

poor health in the household, sacrificing their education, careers or income activities to care for the sick as this is often considered their responsibility [32]. Deep entrenched cultural practices that assign older women the task of caring for the sick and handling deceased bodies introduce another layer of risk [33].

Both men and women are the most affected when it comes to poor health in their household, and without access to media and information, both run a higher risk of being infected with contagious diseases. The presence of radios is an opportunity to tap into for the communities to access more information about zoonotic diseases. In this study, we identify potential channels for health promotion and disease prevention. Community radios, given their widespread usage, surface as effective tools for disseminating health-related updates and education [17]. Similarly, opinion leaders, once sensitized, can serve as crucial links in local health surveillance and reporting networks, highlighting the signs, symptomatic features, and preventive measures for zoonotic diseases in Bundibugyo district [34]. Leaders such as parish chiefs and village health teams are instrumental in detection of cases through local surveillance and reporting of any suspected cases of zoonotic diseases. Opinion leaders also provide ground surveillance when sensitized about the signs, symptoms, transmission mechanisms and prevention measures of zoonotic diseases in their communities [27].

Threats were more skewed to men than women. The threat portion brings to the forefront the potential risks posed by cultural and traditional practices. Roles typically assigned to men, such as animal grazing and hunting, amplify their exposure to zoonotic infections. In most cultures of Uganda, grazing is usually done by men because of their perceived strength and ability to handle animals. Interaction with animals on the other hand increases exposure of men to pathogens of animal origin from their animals or/and from wild animals met during grazing, a potential contributing factor to zoonotic outbreaks [2]. Men in Bundibugyo district are the bread winners of the family, and it is their responsibility to provide food hence involving in very many activities that bring food to the table. Such activities include but are not limited to hunting and the risks associated are not mitigated during these activities. Men are therefore not keen to notice the edible and safe bat species but hunt and eat what they can access, which is a threat to their health. Hence, hunting exposes men and male children to pathogens from the hunted animals some of which could be zoonotic and fatal.

## Conclusion

This study underscores the need for gender mainstreaming as a pivotal lens to understand the diverse impacts of zoonotic diseases, aiming to guide gender-oriented prevention and control interventions. Men are more at risk compared to women, possibly because of their way of life and health-seeking behaviour. A complex web of marginalization, poverty and other vulnerabilities determine a certain lack of control for many women and men over their life and health. It pleads for strategies that acknowledge and incorporate these gender-based strengths, weaknesses, opportunities, and threats to protect all women, men, girls, and boys from the crippling effects of zoonotic diseases in Bundibugyo district, Uganda. In different socio-economic, racial and cultural contexts the situation may vary greatly, and these differences need to be taken into consideration for the development of localized analysis and the resulting solutions.

It captures a snapshot of our current situation, ultimately aiming to inform policies and strategies that ensure a healthier future. In addition, addressing these issues requires a thorough understanding of the gender dynamics at play, active health education programs, and enabling interventions tailored to the needs of these communities. Doing so would ensure that individuals, regardless of their gender, are equipped with the necessary knowledge to navigate through zoonotic diseases and other related health challenges.

## Supporting information

**S1 Data. Consent forms for data collection.**
(PDF)

**S1 File. Qualitative tool for Key Informants, In-depth interviews and Focus Group Discussions.**
(DOCX)

## Acknowledgments

The authors wish to thank the District Veterinary Team and District Health Team of Bundibugyo district, the participants of the study.

## Author contributions

**Conceptualization:** Clovice Kankya, James Muleme, Lydia Nabawanuka Namakula, George Seruwagi, Christine Mbabazi Mpyangu, Lesley Rose Ninsiima.

**Data curation:** Clovice Kankya, James Muleme, Lydia Nabawanuka Namakula, George Seruwagi, Christine Mbabazi Mpyangu, Lesley Rose Ninsiima.

**Formal analysis:** Clovice Kankya, James Muleme, Lydia Nabawanuka Namakula, Christine Mbabazi Mpyangu, Lesley Rose Ninsiima.

**Funding acquisition:** Clovice Kankya, James Muleme.

**Investigation:** Clovice Kankya, Lesley Rose Ninsiima.

**Methodology:** Clovice Kankya, James Muleme, Lydia Nabawanuka Namakula, George Seruwagi, Christine Mbabazi Mpyangu, Lesley Rose Ninsiima.

**Resources:** Christine Mbabazi Mpyangu.

**Supervision:** Lesley Rose Ninsiima.

**Validation:** Clovice Kankya, Lydia Nabawanuka Namakula, George Seruwagi, Lesley Rose Ninsiima.

**Visualization:** Clovice Kankya, James Muleme, Lydia Nabawanuka Namakula, Christine Mbabazi Mpyangu, Lesley Rose Ninsiima.

**Writing – original draft:** Clovice Kankya, James Muleme, Lydia Nabawanuka Namakula, George Seruwagi, Christine Mbabazi Mpyangu, Lesley Rose Ninsiima.

**Writing – review & editing:** Clovice Kankya, James Muleme, Lydia Nabawanuka Namakula, Christine Mbabazi Mpyangu, Lesley Rose Ninsiima.

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
