## [Decision Letter · Decision Letter 0]

10 Jul 2024

PONE-D-24-21886Gender perspectives on zoonotic disease epidemiology; A Strength Weakness Opportunities Threats analysis in Bundibugyo district, Uganda.PLOS ONE

Dear Dr. Kankya,

Thank you for submitting your manuscript to PLOS ONE. After careful consideration, we feel that it has merit but does not fully meet PLOS ONE’s publication criteria as it currently stands. Therefore, we invite you to submit a revised version of the manuscript that addresses the points raised during the review process.

**Additional Editor Comments:**

**In addition to specific comments from three reviewers, please:**

**- Modify the figure 1. I advise you to use freely accessible layers and Geographic Information System software to generate your own map showing precise study sites.**

- Line 202-203: Where is the final list of prioritized zoonoses

- 375 – 387: What is the relevance of this paragraph to your study findings?

- Check for the manuscript for the many typos. Examples are:

o Line 125: Missing word in the sentence ending with “Lake Albert to the…”

o Line 126: The sentence is incomplete

o 196-197: Themes c and d are the same.

o Line 202: Is Swine fever a zoonoses? Clarify why it was included

o 306-307: “Importance of” is repeated twice

We look forward to receiving your revised manuscript.

Kind regards,

Anselme Shyaka, Ph.D

Academic Editor

PLOS ONE

Journal Requirements:

We acknowledge funding obtained from Norwegian Agency for Development Cooperation (NORAD) through the NORHED-II Program and the project Climate Change and Infectious Diseases - A One Health Approach (CIDIMOH), grant number 68802. 

The authors wish to thank the District Veterinary Team and District Health Team of Bundibugyo district, the participants of the study and funders of this study.

We acknowledge funding obtained from Norwegian Agency for Development Cooperation (NORAD) through the NORHED-II Program and the project Climate Change and Infectious Diseases - A One Health Approach (CIDIMOH), grant number 68802. 

6. We note that you have referenced "S. N. Musila, D. K. Mafigiri and M. Schmidt-sane" which has currently not yet been accepted for publication. Please remove this from your References and amend this to state in the body of your manuscript: (S. N. Musila, D. K. Mafigiri and M. Schmidt-sane [Submitted]) as detailed online in our guide for authors

7. We note that Figure 1 in your submission contain map images which may be copyrighted. All PLOS content is published under the Creative Commons Attribution License (CC BY 4.0), which means that the manuscript, images, and Supporting Information files will be freely available online, and any third party is permitted to access, download, copy, distribute, and use these materials in any way, even commercially, with proper attribution. For these reasons, we cannot publish previously copyrighted maps or satellite images created using proprietary data, such as Google software (Google Maps, Street View, and Earth). For more information, see our copyright guidelines: http://journals.plos.org/plosone/s/licenses-and-copyright.

We require you to either present written permission from the copyright holder to publish these figures specifically under the CC BY 4.0 license, or remove the figures from your submission:

Reviewers' comments:

Reviewer's Responses to Questions

**Comments to the Author**

1. Is the manuscript technically sound, and do the data support the conclusions?

Reviewer #1: Yes

Reviewer #2: Yes

Reviewer #3: Yes

2. Has the statistical analysis been performed appropriately and rigorously? 

Reviewer #1: N/A

Reviewer #2: N/A

Reviewer #3: N/A

3. Have the authors made all data underlying the findings in their manuscript fully available?

Reviewer #1: Yes

Reviewer #2: Yes

Reviewer #3: Yes

4. Is the manuscript presented in an intelligible fashion and written in standard English?

Reviewer #1: Yes

Reviewer #2: Yes

Reviewer #3: Yes

5. Review Comments to the Author

**Reviewer #1:**  Authors addressed clearly aspects related on the Gender perspectives on zoonotic disease epidemiology using a SWOT analysis in Bundibugyo a district of Uganda. The methodology is fitting the results presented in the manuscript and the topic is actually relevant for the control of zoonotic diseases in the country an the sub-region.

Therefore some comments are proposed to improve this manuscript:

>Line 73: Better to put a citation (s) for this sentence

>Line 76: remove the coma at the beginning of the sentence

>Line 87: In my opinion, not all men are playing such role except where this gender role can be played, so reformulate the sentence.

>Line 137: Is it possible to make clear the district of study by extracting it from the map of Uganda. There the borders could be well visible.

>147: make a correction on this sentence : All participants mostly understood English.

How about to include also participants who didn't get a formal education; because zoonotic diseases mitigation can not be limited to knowledge. Illiteracy can maybe play a role of their spread and therefore constitute a weakness for their control. What do you think on this?

>Line 288: diseases

>Line 310: In this paragraph, the capture on how diseases manifest and traverse along lines of gender is not clearly stated.

Also, Check if some of the opportunities although mentioned by participants are not likely to be part of the Strength for zoonotic diseases management, prevention and control

>Line 435: remove the citation in the conclusion

**Reviewer #2:**  - there were some unclear sentences with some run-ons, that i've noted in the attachment. Suggestions on improving sentence construction were also made to clarify some statements

- examples of questions from the IDI, KII, and FGD would be helpful, if authors will oblige

- suggest for the authors to proofread the entire report again, as the abstract has an obvious mistake (opportunities mentioned twice, no threats)

**Reviewer #3: ** The study explores gender perspectives on zoonotic disease epidemiology

using a (strength, weakness, opportunities and threats) SWOT analysis in Bundibugyo district in Uganda.

The study adopts the SWOT analysis. While the authors have made good effort to do so across the key elements, there is need for further clarity within each SWOT element in the discussion session. This should be done by highlighting the gendered aspects in each element e.g. in discussing a strength, discuss how it is a strength for men, for women and why.

The authors should also look within the uploaded document and address specific comments mentioned.

6. PLOS authors have the option to publish the peer review history of their article (what does this mean? ). If published, this will include your full peer review and any attached files.

**Do you want your identity to be public for this peer review?** For information about this choice, including consent withdrawal, please see our Privacy Policy .

Reviewer #1: No

Reviewer #2: No

Reviewer #3: **Yes: ** Esther Leah Achandi

---

## [Author Response · Author response to Decision Letter 1]

29 Aug 2024

Makerere University

School of biosecurity, ecosystems and veterinary Public Health

26th July 2024

To The Editor

Plos One

We are grateful to the editor for giving us the opportunity to edit our manuscript titled “Gender perspectives on zoonotic disease epidemiology; A Strength Weakness Opportunities Threats analysis in Bundibugyo district, Uganda.” We also appreciate the peer reviewers’ efforts for their insights, which have significantly improved the quality of our manuscript.

We have addressed the reviewers’ comments to the best of our ability, hence would like to submit the revised manuscript.

Comment Response

Editor

Modify the figure 1. I advise you to use freely accessible layers and Geographic Information System software to generate your own map showing precise study sites. This has been edited to highlight the district of study

Line 202-203: Where is the final list of prioritized zoonoses All zoonoses highlighted were mentioned by participants as diseases that are of priority to them.

375 – 387: What is the relevance of this paragraph to your study findings? This shows the gender roles men and women play in prevention of zoonotic diseases from both animals and humans and how exposure to this knowledge has helped them.

Check for the manuscript for the many typos. Examples are:

o Line 125: Missing word in the sentence ending with “Lake Albert to the…” Thank you for this comment. This has been edited as per the comment given.

Line 126: The sentence is incomplete The sentence has been completed.

196-197: Themes c and d are the same.

This has been edited to include community strength for zoonotic disease management as theme c

Line 202: Is Swine fever a zoonoses? Clarify why it was included This has been removed as per the comment since it’s not a zoonotic disease.

306-307: “Importance of” is repeated twice Thank you for this comment. This has been edited.

Reviewer 1

Authors addressed clearly aspects related on the Gender perspectives on zoonotic disease epidemiology using a SWOT analysis in Bundibugyo a district of Uganda. The methodology is fitting the results presented in the manuscript and the topic is actually relevant for the control of zoonotic diseases in the country an the sub-region.

Therefore some comments are proposed to improve this manuscript:

>Line 73: Better to put a citation (s) for this sentence The citation has been included. Thank you for the comment.

>Line 76: remove the coma at the beginning of the sentence This has been removed as per the comment. Thank you for the comment.

>Line 87: In my opinion, not all men are playing such role except where this gender role can be played, so reformulate the sentence. Thank you for the comment. This statement has been reformulated.

>Line 137: Is it possible to make clear the district of study by extracting it from the map of Uganda. There the borders could be well visible. Thank you for the comment. The map has been deleted form this article.

>147: make a correction on this sentence: All participants mostly understood English. This has been edited as per the comment given.

How about to include also participants who didn't get a formal education, because zoonotic diseases mitigation cannot be limited to knowledge. Illiteracy can maybe play a role of their spread and therefore constitute a weakness for their control. What do you think on this? Thank you for this comment. Most participants understand selected for this study were found to understand English as well however originally during the selection knowing English was not a selection criterion.

>Line 288: diseases Thank you for the comment. This has been edited to include zoonotic diseases.

>Line 310: In this paragraph, the capture on how diseases manifest and traverse along lines of gender is not clearly stated.

Thanks for the comment. This quote was aimed to show how the women prevent the diseases by having the prevention strategies in place to prevent the spread of the zoonotic/infectious diseases. I see gender inclusiveness through prevention from homes.

Also, Check if some of the opportunities although mentioned by participants are not likely to be part of the Strength for zoonotic diseases management, prevention and control The strength for zoonotic disease management, prevention and control is looking at what are the advantages and what is working well within the community. Opportunities for communities in management leaveraged on what prizes are within reach for the community to dwell on.

>Line 435: remove the citation in the conclusion Thank you for the comment. This has been removed.

Reviewer 2

There were some unclear sentences with some run-ons, that i've noted in the attachment. Suggestions on improving sentence construction were also made to clarify some statements

- examples of questions from the IDI, KII, and FGD would be helpful, if authors will oblige

Thank you for the comment. The tool has been uploaded.

- suggest for the authors to proofread the entire report again, as the abstract has an obvious mistake (opportunities mentioned twice, no threats) Thanks for the comment. The repetition has been edited and all included.

Reviewer 3

The study explores gender perspectives on zoonotic disease epidemiology

using a (strength, weakness, opportunities and threats) SWOT analysis in Bundibugyo district in Uganda. The study adopts the SWOT analysis. While the authors have made a good effort to do so across the key elements, there is need for further clarity within each SWOT element in the discussion session. This should be done by highlighting the gendered aspects in each element e.g. in discussing a strength, discuss how it is a strength for men, for women and why.

The authors should also look within the uploaded document and address specific comments mentioned. Thank you for the comment. This has been highlighted in the whole discussion how it is gender related within each SWOT. All comments have been addressed and worked on.

Yours Sincerely

Clovice Kankya

---

## [Decision Letter · Decision Letter 1]

28 Mar 2025

PONE-D-24-21886R1Gender perspectives on zoonotic disease epidemiology; A Strength Weakness Opportunities Threats analysis in Bundibugyo district, Uganda.PLOS ONE

Dear Dr. Kankya,

Thank you for submitting your manuscript to PLOS ONE. After careful consideration, we feel that it has merit but does not fully meet PLOS ONE’s publication criteria as it currently stands. Therefore, we invite you to submit a revised version of the manuscript that addresses the points raised during the review process.

We look forward to receiving your revised manuscript.

Kind regards,

Ayi Vandi Kwaghe, D.V.M., M.V.Sc., P.G.D.E. Ph.D., MPH, FETP

Academic Editor

PLOS ONE

Journal Requirements:

Reviewers' comments:

Reviewer's Responses to Questions

**Comments to the Author**

1. If the authors have adequately addressed your comments raised in a previous round of review and you feel that this manuscript is now acceptable for publication, you may indicate that here to bypass the “Comments to the Author” section, enter your conflict of interest statement in the “Confidential to Editor” section, and submit your "Accept" recommendation.

Reviewer #1: All comments have been addressed

Reviewer #4: All comments have been addressed

2. Is the manuscript technically sound, and do the data support the conclusions?

Reviewer #1: Yes

Reviewer #4: Yes

3. Has the statistical analysis been performed appropriately and rigorously? 

Reviewer #1: N/A

Reviewer #4: (No Response)

4. Have the authors made all data underlying the findings in their manuscript fully available?

Reviewer #1: Yes

Reviewer #4: Yes

5. Is the manuscript presented in an intelligible fashion and written in standard English?

Reviewer #1: Yes

Reviewer #4: Yes

6. Review Comments to the Author

Reviewer #1: Authors touched a topic addressing the Gender perspectives on zoonotic disease epidemiology; A Strength Weakness

Opportunities Threats analysis in Bundibugyo district, Uganda. They responded correctly all the comments from the reviewers.

Reviewer #4: The authors of the manuscript entitled: Gender perspectives on zoonotic disease epidemiology; A Strength Weakness Opportunities Threats analysis in Bundibugyo district, Uganda, have improved the presentation of the work by addressing all comments from reviewers.

This paper would have been beneficial if a list of zoonotic diseases mentioned by the respondents are provided before the ranking?. Whereas SWOT approach was used, it not very clear how it was deployed across KIIs, Indepth interviews and FGDs (This could be provided as supplementary files including checklists that were used). It was not clear on who conducted different interviews, and which language was used.

In the methods sections, 7 sub-counties were considered in the study area, it is not clear on how the respondents for different categories of interviews were selected in 7- sub-counties (A supplementary table) may be need to clarify the approaches used in the selection of respondents.

Some sections in the discussion seem to divert from the subject matter and introduce most the previously reported assertions

7. PLOS authors have the option to publish the peer review history of their article (what does this mean? ). If published, this will include your full peer review and any attached files.

**Do you want your identity to be public for this peer review?** For information about this choice, including consent withdrawal, please see our Privacy Policy .

Reviewer #1: No

Reviewer #4: **Yes: ** Lawrence Mugisha

---

## [Author Response · Author response to Decision Letter 2]

10 Apr 2025

Makerere University

School of biosecurity, ecosystems and veterinary Public Health

10th April 2024

To The Editor

Plos One

We are grateful to the editor for giving us the opportunity to edit our manuscript titled “Gender perspectives on zoonotic disease epidemiology: A Strength Weakness Opportunities Threats analysis in Bundibugyo district, Uganda.” We also appreciate the peer reviewers’ efforts for their insights, which have significantly improved the quality of our manuscript.

We have addressed the reviewers’ comments to the best of our ability, hence would like to submit the revised manuscript.

Comment Response

Reviewer #1

Authors touched a topic addressing the Gender perspectives on zoonotic disease epidemiology; A Strength Weakness

Opportunities Threats analysis in Bundibugyo district, Uganda. They responded correctly all the comments from the reviewers.

Thank you for the feedback, we are really grateful.

Reviewer #4

The authors of the manuscript entitled: Gender perspectives on zoonotic disease epidemiology; A Strength Weakness Opportunities Threats analysis in Bundibugyo district, Uganda, have improved the presentation of the work by addressing all comments from reviewers.

This paper would have been beneficial if a list of zoonotic diseases mentioned by the respondents are provided before the ranking?

Thank you for the comment, these have been included in the manuscript.

Whereas SWOT approach was used, it not very clear how it was deployed across KIIs, In-depth interviews and FGDs (This could be provided as supplementary files including checklists that were used). It was not clear on who conducted different interviews, and which language was used.

Thank you for the comment. The tool has been included a supplementary material for each of the categories of KIIs, IDIs and FGDs.

In the methods sections, 7 sub-counties were considered in the study area, it is not clear on how the respondents for different categories of interviews were selected in 7- sub-counties (A supplementary table) may be need to clarify the approaches used in the selection of respondents. This is a great comment. The participants recruitment criteria were already explained in details in the manuscript.

Some sections in the discussion seem to divert from the subject matter and introduce most the previously reported assertions

Thanks for the comment. These have been further read and edited

Yours Sincerely

Clovice Kankya

---

## [Editor Report · Decision Letter 2]

25 Apr 2025

Gender perspectives on zoonotic disease epidemiology; A Strength Weakness Opportunities Threats analysis in Bundibugyo district, Uganda.

PONE-D-24-21886R2

Dear Dr. Kankya,

We’re pleased to inform you that your manuscript has been judged scientifically suitable for publication and will be formally accepted for publication once it meets all outstanding technical requirements.

Kind regards,

Ayi Vandi Kwaghe, D.V.M., M.V.Sc., P.G.D.E. Ph.D., MPH, FETP

Academic Editor

PLOS ONE

Additional Editor Comments (optional): All comments were addressed by the authors.
---

## [Editor Report · Acceptance letter]

PONE-D-24-21886R2

PLOS ONE

Dear Dr. Kankya,

I'm pleased to inform you that your manuscript has been deemed suitable for publication in PLOS ONE. Congratulations! Your manuscript is now being handed over to our production team.

Kind regards,

on behalf of

Dr. Ayi Vandi Kwaghe

Academic Editor

PLOS ONE